# Differential Effects of Tree Species on Soil Microbiota 45 Years after Afforestation of Former Pastures

Richard Gere [ID], Mikuláš Kočiš, Ján Židó, Dušan Gömöry [ID] and Erika Gömöryová *[ID]

Faculty of Forestry, Technical University in Zvolen, TG Masaryka 24, SK-96001 Zvolen, Slovakia;
r.gere60@gmail.com (R.G.); kocis@is.tuzvo.sk (M.K.); xzido@is.tuzvo.sk (J.Ž.); gomory@tuzvo.sk (D.G.)
* Correspondence: gomoryova@tuzvo.sk

**Abstract:** Several decades ago, many former pastures in Central Europe were afforested or colonized by trees after being abandoned. Knowledge of the effects of tree species on soil properties is important for planning of the composition of future forests. In this regard, a research location in Vrchdobroč (Central Slovakia), which is former agricultural land used as pasture, enables the exploration of ecosystem processes and properties in stands of different tree species after afforestation. The goal of our study was to find out whether changes in soil properties, including soil microbial activity and diversity among different stands, were already observable 45 years after the afforestation, and how the effects differed among stands of different tree species. The study was conducted in the pure stands of Norway spruce (*Picea abies* L. Karst.), Douglas fir (*Pseudotsuga menziesi* (Mirb.) Franco), European beech (*Fagus sylvatica* L.) and sycamore maple (*Acer pseudoplatanus* L.). Multivariate analyses of physico-chemical soil properties indicated an overlap between the soils under the Douglas fir and the spruce, but a clear separation of beech from sycamore. In general, both microbial activity and diversity were, surprisingly, highest under the Douglas fir, followed by the sycamore, with the beech and the spruce showing mostly lower values.

**Keywords:** Norway spruce; Douglas fir; beech; maple; soil chemical properties; soil microorganisms



## 1. Introduction

Soil and plants are linked very tightly. Soil properties affect plant growth and, conversely, plants affect soil attributes. The effect of plants on soil properties is very distinct, especially in forest ecosystems because of the long-term influence of forest stand on soil. Trees can affect soil properties directly through the input of organic material (dead organic matter, root exudates), living tissues (roots) and/or indirectly via modification of microclimate, e.g., radiation reaching the ground, evaporation from the soil surface, relative air humidity, air circulation, water input to the soil, etc., [1–5]. The effects differ depending on tree species, because of the different qualities and quantities of organic residues left on the soil surface or going directly into the soil, as well as different crown and root architecture, canopy openness, stemflow rate, etc., [5–7]. Numerous studies have shown that differences in tree cover are reflected especially in the thickness of the surface organic layer, in the soil acidity, in the base saturation, and in the carbon and nitrogen concentrations, and consequently in the responses of the soil microbial biomass, the activity and structure, and the diversity of the microbial communities [8–10].

Human activity has often caused decreases in forest areas—especially in favor of agriculture—in many parts of the world, and deforestation has persisted. In Europe, the minimum of forest cover occurred during the 18th and 19th centuries [1]. Increasing demand for wood production led to the planting and extensive use of conifers in many European countries, often at the expense of native deciduous tree species. Conifers and deciduous trees are traditionally considered to differ in their effect on soil properties, especially because of their different qualities of litter input which, in the case of conifers,

can lead to soil acidification, deterioration and podzolization in the long term [11,12]. Nevertheless, there are also contradictory results indicating similar or even better soil properties under certain conifer species in comparison to broadleaves [13,14]. According to a meta-analysis by Augusto, et al. [1], the discrepancy can be partially explained by variation between the soils of some of the study sites; therefore, they emphasized the importance of experimental design. As they stated, there are only a few sufficiently replicated studies with the same stand age and stand management, on the same soil type and with the same land use history.

In Slovakia, the locality of Vrchdobroč is former agricultural land which was used for decades as pasture. In 1960, the area, comprising 26,000 ha, was declared important for water management, as it contained the source of two rivers. During 1960–1985, an area of over 5166 ha was afforested to increase the forest cover from 29 to 49%. Within this area, 283 ha, afforested by 17 tree species, was used for research and demonstration purposes, managed by the Forestry Research Institute of the National Forestry Centre in Zvolen (Slovakia). Some of the plots were damaged by drought, animals, snow, etc., and nowadays conifers and broadleaves represent 85% and 15% of the area, respectively [15].

The research location allows for the exploration of ecosystem processes and properties in stands of different tree species after afforestation. In this study, we used it to evaluate the impact of different tree species on soil, with emphasis on chemical and microbial properties. We focused on pure forest stands of two coniferous and two deciduous species of the same age and management history, growing under almost identical environmental conditions (soil type, altitude, climate, etc.). Our goal was to find out whether changes in soil properties among different stands were already observable 45 years after the afforestation; what had been the respective effects of the different tree species; and whether those effects had also been reflected in the properties of the microbial community (biomass, activity, diversity). We hypothesized that 45 years was sufficiently long enough for the effects of trees on soil properties to be manifested, and that changes would have occurred especially in the surface organic layer, which started to form after afforestation, and probably also in the top 10 cm of the mineral soil horizons. As the quality of the litter of conifers essentially differs from that of broadleaves, we expected differences to be especially notable between these two groups of trees.

## 2. Materials and Methods

### 2.1. Study Sites and Soil Sampling

The study area, Vrchdobroč, is located in the Veporské vrchy Mts., Central Slovakia. The mean annual temperature is 5 °C; mean temperature in July 15 °C; and in January −6 °C. The yearly precipitation reaches 920 mm. The dominant soil type is Cambisol with a sandy loam texture formed from porphyric granodiorites and granites. The soil skeleton content is 20–50%.

Pure 45-year-old stands of Norway spruce (*Picea abies* (L.) Karst.), Douglas fir (*Pseudotsuga menziesi* (Mirb.) Franco), European beech (*Fagus sylvatica* L.) and sycamore maple (*Acer pseudoplatanus* L.), randomly distributed across the area, were selected for the study. Three stands of each species were chosen. The stand area of each tree species varied between 0.40 and 1.59 ha. The plots were situated at the altitudes of 815–850 m a. s. l., with S-SE aspect, on a slope of 5–10°. The stands were managed with sylvicultural methods, which are traditional in the Carpathian mountain forests: conifer stands underwent cleaning every 20 years, and thinning from below at around 35 years; broadleaved stands were cleaned twice (at 15 and 25 years), and subsequently thinned from above at the age of 40 years [15].

Five soil samples were collected along linear transects, with 10 m spacing in each stand, in July 2015. Samples from the O-horizon were collected, using a 0.2 m × 0.2 m frame placed on the soil surface, while the humus layer underneath the template was cut by knife from its surroundings. After removing the surface organic layer, mineral soil samples were taken from a depth of 0–0.1 m (the A-horizon), using the knife and shovel with a

depth scale indication. We did not use a probe sampler because of high skeleton content in the soil. Approximately 400 g of soil samples were put into plastic bags, i.e., a comparable soil weight from all sampling plots. Visible coarse particles (e.g., roots, fauna, stones) were removed. Samples from deeper layers were not taken because we assumed that the effect of the trees would be notable, especially in the topsoil horizons. In total, 120 soil samples were collected (4 tree species × 3 stands × 2 horizons × 5 replications).

### 2.2. Laboratory Analyses

After bringing the samples to the laboratory, they were divided into two parts. One part, intended for microbial analyses, was stored in a refrigerator. The second part was air-dried immediately after being brought in, and analyzed for chemical properties. For the determination of soil moisture and dry weight, a gravimetric method was used, based on oven-drying of fresh soil samples from the O-horizon at 60 °C and from the A-horizon at 105 °C until a constant weight. Soil acidity (pH-$CaCl_2$) was determined potentiometrically in suspension prepared from 2.5 g of litter or 10 g of mineral soil and 25 mL of 0.01 M $CaCl_2$. The C, N and S concentrations were measured using a VarioMacro Elemental Analyzer (CNS Version, Elementar Gmbh, Langenselbold, Germany), employing the dry combustion method. Exchangeable cations of $Ca^{2+}$, $Mg^{2+}$ and $K^+$ were determined in 1 M extract of $NH_4Cl$, using atomic absorption spectrometry (GBC Avanta AAS, Dandenong, Victoria, Australia), and evaluated only for samples from the mineral A-horizon.

Determination of basal respiration (BR) was performed according to Isermeyer's method [16]. The amount of $CO_2$ released from a fresh soil sample in a glass container for 24 h and absorbed in 0.05 M NaOH was measured. The amount of carbonate was determined by titration with 0.05 M HCl after 5 mL of $BaCl_2$ and phenolphthalein were added. For the determination of substrate-induced respiration (SIR), 0.5 g and 0.125 g of glucose was added to samples from the A-horizon and the O-horizon, respectively. The evolved $CO_2$ was measured, as described above, after 4.5 h [16]. Catalase activity (Cat) was estimated according to the method described by Khaziev [17], based on the measurement of the released $O_2$, 10 min after 3% $H_2O_2$ was added to a fresh soil sample. Microbial biomass carbon (Cmic) was determined using the microwave-irradiation procedure [18]. C concentration was quantified titrimetrically, after the oxidation of the extract with $K_2Cr_2O_7$/$H_2SO_4$. N-mineralization (Nmin) was measured according to the procedure described by Kandeler [19]. Soil samples under anaerobic conditions were incubated at 40 °C for 7 days, and the released $NH_4$-N was estimated by a colorimetric procedure. Nmin was determined only in the samples from the A-horizon.

The Biolog® method was used to determine the activity of the microbial functional groups [20]. 0.5 g of soil sample was placed into a plastic bank. After the addition of 50 mL 0.85% NaCl solution, the suspension was left for 45 min on automatic shaker, and then filtered. Subsequently, the supernatant was diluted to 1:1000 and 1:10,000 for samples from the O- and A-horizon, respectively. 150 μL of the extract was pipetted into BIOLOG Ecoplates, and incubated at 27 °C for 5 days. Absorbance at 590 nm was measured spectrophotometrically using a Sunrise Microplate reader (Tecan, Salzburg, Austria) every 24 h. Absorbance values were blanked against the control well. The metabolic activity was calculated as the area below the time–absorbance curve, and was used as a measure of the abundance of the respective functional group. The richness of the soil functional groups was assessed as the number of substrates with a non-zero response. The functional diversity of the microbial community was assessed by Hill's diversity index ($N_2$) Equation (1) [21]:

$$N_2 = 1/\sum \mathrm{p}i^2 \tag{1}$$

where p$i$ is the ratio of the activity on a particular substrate to the sum of activities on all substrates.

### 2.3. Statistical Analysis

The effect of different tree species on soil and microbial properties was evaluated by one-way ANOVA (tree species being considered a fixed-effect factor), with subsequent Tukey's HSD post-hoc tests conducted separately for the O- and A-horizons, using Statistica 12 software [22].

As microbial richness and diversity do not completely explain how the structure of community changes with stand-forming trees species, a multivariate analysis (canonical correspondence analysis; CCA) was performed using CANOCO 5 software [23]. Analyses were done separately for physico-chemical properties, microbial community parameters and community-level physiological profiles assessed by the Biolog® method.

## 3. Results

### 3.1. Soil Chemical Properties

We observed significant differences in the effects of tree species on soil chemical properties; however, the effects differed between soil horizons (Table 1). CCA ordination diagrams (Figure 1), based on physico-chemical properties, clearly show the distinction between sycamore maple, beech and coniferous stands. No significant differences were found in soil chemical characteristics between the spruce and the Douglas fir stands in both horizons; however, litter weight was higher in spruce stands. The differences between the beech and sycamore stands in the O-horizon were less pronounced than in the A-horizon: generally, chemical soil properties (SOC, nutrients, base saturation) were more favorable in sycamore stands. Nevertheless, no general difference in favor of broadleaves compared to conifers was observed.

**Table 1.** Basic statistics (means ± standard deviations) of physico-chemical soil properties in stands of different tree species.

| Physico-Chemical Properties | Horizon | P | *Picea* | *Pseudotsuga* | *Fagus* | *Acer* |
|---|---|---|---|---|---|---|
| pH/CaCl$_2$ | O | <0.1 | 4.85 ± 0.40 a | 5.08 ± 0.35 a | 5.06 ± 0.21 a | 5.06 ± 0.35 a |
| | A | <0.001 | 3.73 ± 0.09 b | 3.94 ± 0.28 b | 3.89 ± 0.23 b | 4.28 ± 0.26 a |
| Soil moisture (%) | O | <0.01 | 29.71 ± 6.90 c | 61.62 ± 54.34 a | 32.47 ± 12.37 bc | 37.63 ± 13.76 b |
| | A | <0.001 | 17.10 ± 3.12 b | 15.29 ± 5.52 b | 23.28 ± 7.22 ab | 36.80 ± 10.18 a |
| C (%) | O | <0.001 | 32.37 ± 10.05 a | 32.53 ± 8.64 a | 22.61 ± 7.79 b | 21.42 ± 5.01 b |
| | A | <0.001 | 4.60 ± 0.85 b | 4.15 ± 1.05 b | 3.73 ± 0.71 b | 7.19 ± 1.07 a |
| N (%) | O | <0.001 | 1.65 ± 0.48 a | 1.70 ± 0.40 a | 1.22 ± 0.35 b | 1.48 ± 0.25 ab |
| | A | <0.001 | 0.37 ± 0.06 b | 0.35 ± 0.08 b | 0.34 ± 0.05 b | 0.67 ± 0.09 a |
| S (%) | O | 0.92 | 0.24 ± 0.06 a | 0.24 ± 0.05 a | 0.23 ± 0.23 a | 0.21 ± 0.05 a |
| | A | <0.001 | 0.07 ± 0.03 b | 0.07 ± 0.02 b | 0.06 ± 0.01 b | 0.10 ± 0.02 a |
| C:N ratio | O | <0.001 | 19.40 ± 1.41 a | 18.97 ± 2.07 a | 18.19 ± 1.46 a | 14.37 ± 1.05 b |
| | A | <0.001 | 12.46 ± 0.92 a | 11.87 ± 1.02 a | 10.81 ± 0.67 b | 10.82 ± 0.70 b |
| Litter weight (kg·m$^{-2}$) | O | <0.001 | 1.46 ± 0.36 a | 1.24 ± 0.31 b | 0.85 ± 0.27 b | 1.45 ± 0.44 a |
| Ca$^{2+}$ (mg·kg$^{-1}$) | A | <0.001 | 475.1 ± 168.6 b | 742.8 ± 413.1 b | 692.7 ± 163.5 b | 1272.5 ± 379.5 a |
| Mg$^{2+}$ (mg·kg$^{-1}$) | A | <0.001 | 116.0 ± 33.9 b | 131.2 ± 21.9 b | 135.6 ± 13.3 b | 178.3 ± 29.8 a |
| K$^+$ (mg·kg$^{-1}$) | A | <0.001 | 49.13 ± 6.37 b | 56.11 ± 36.41 b | 67.23 ± 15.61 b | 86.99 ± 27.51 a |

*P*—probability associated with ANOVA *F*-test for tree species effect; different letters designate homogeneous groups based on Tukey's HSD post-hoc tests. O—forest floor; A—mineral horizon.

The O-horizon exhibited lower soil acidity, higher C/N ratio and higher concentration of C, N, S than the A-horizon. Generally, higher decline of C and N content from the O- to the A-horizon was observed under conifers in comparison to deciduous stands. Soil pH of the O-horizon, surprisingly, did not differ significantly among tree species. On the other hand, significantly higher C concentration was found in the litter of conifers than the deciduous trees. Beech litter exhibited the lowest N content, and the litter of maple the lowest C/N ratio. However, in the A-horizon the pattern differed. Soils in the maple stands showed the lowest acidity and the highest content of all nutrients, in comparison to other stands. Surprisingly, soil properties under beech stands did not differ significantly

from those under the conifers. The C/N ratio was the only parameter that differed between the two groups —conifers and deciduous stands.

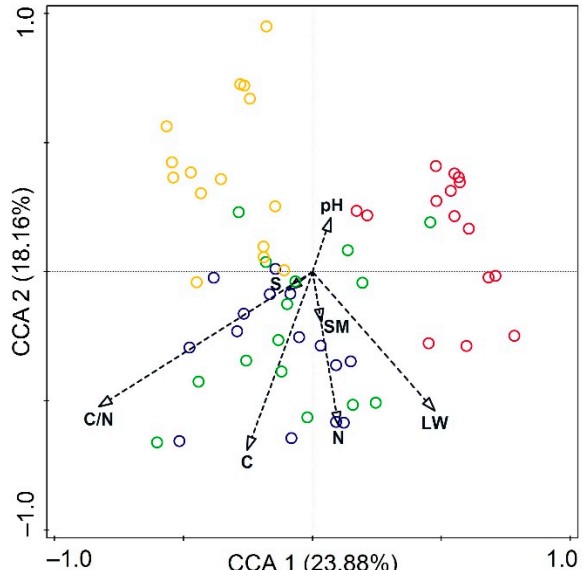 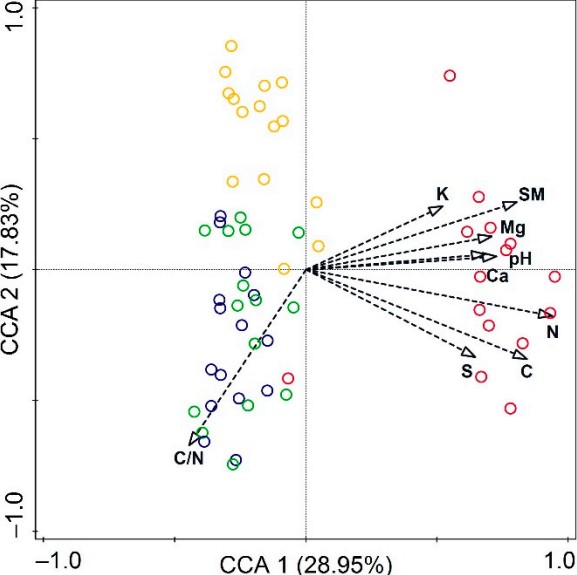

**Figure 1.** Canonical correspondence analysis for physico-chemical soil properties and different forest stands within the O-horizon (**left**) and the A-horizon (**right**). Arrows represent the direction of the steepest increase of individual physico-chemical properties. Circles represent individual sampling sites, with color distinctions of dominant tree species in the stands (blue—*Pseudotsuga menziesii*; green—*Picea abies*; yellow—*Fagus sylvatica*; red—*Acer pseudoplatanus*).

*3.2. Soil Microbial Characteristics*

Like chemical properties, soil microbial characteristics also differed significantly between the stands of different tree species (Table 2). However, as shown by the CCA ordination, the pattern of differences did not follow completely that of the soil physico-chemical properties, and especially in the O-horizon, clear differences in microbial characteristics between the conifer stands were obvious (Figure 2). Again, in general, microbial characteristics in deciduous stands did not significantly differ from those of the conifer stands.

**Table 2.** Basic statistics (means ± standard deviations) of soil microbial characteristics in stands of different tree species.

| Microbial Properties | Horizon | *P* | *Picea* | *Pseudotsuga* | *Fagus* | *Acer* |
|---|---|---|---|---|---|---|
| Basal respiration | O | <0.001 | 1.37 ± 0.49 b | 4.47 ± 2.89 a | 1.88 ± 1.51 b | 2.61 ± 2.38 ab |
| ($\mu g\ CO_2 \cdot g^{-1} \cdot h^{-1}$) | A | 0.11 | 0.18 ± 0.04 a | 0.16 ± 0.09 a | 0.20 ± 0.09 a | 0.42 ± 0.63 a |
| Substrate-induced | O | <0.001 | 5.02 ± 1.97 b | 22.00 ± 23.62 a | 5.37 ± 2.42 b | 9.64 ± 6.67 b |
| respiration ($\mu g\ CO_2 \cdot g^{-1} \cdot h^{-1}$) | A | <0.001 | 1.09 ± 0.61 b | 0.63 ± 0.29 b | 1.11 ± 0.46 b | 2.40 ± 0.82 a |
| Catalase activity | O | <0.05 | 5.26 ± 0.93 b | 6.02 ± 2.26 ab | 6.62 ± 1.26 ab | 6.76 ± 1.39 a |
| ($ml\ O_2 \cdot g^{-1} \cdot min^{-1}$) | A | <0.001 | 0.65 ± 0.20 a | 0.56 ± 0.17 a | 0.62 ± 0.21 a | 1.06 ± 0.26 a |
| N mineralization | A | <0.001 | 0.74 ± 0.50 b | 1.41 ± 1.04 b | 4.00 ± 2.17 a | 5.38 ± 4.36 a |
| ($\mu g\ NH_4^+$-$N \cdot g^{-1} \cdot d^{-1}$) | | | | | | |
| Microbial biomass carbon | O | <0.001 | 6440 ± 1250 ab | 7869 ± 3339 a | 4338 ± 1047 c | 4894 ± 1747 bc |
| ($\mu g \cdot g^{-1}$) | A | <0.001 | 568.2 ± 169.7 b | 420.3 ± 99.2 c | 361.8 ± 105.3 c | 912.4 ± 185.4 a |
| Richness of functional groups | O | <0.001 | 26.87 ± 2.13 c | 28.93 ± 1.79 a | 27.47 ± 1.99 ab | 28.40 ± 1.59 ab |
| | A | <0.001 | 27.80 ± 1.01 a | 26.87 ± 2.13 a | 25.07 ± 1.79 b | 26.93 ± 1.75 a |
| Diversity of functional groups | O | <0.001 | 10.48 ± 2.63 b | 17.60 ± 1.97 a | 16.55 ± 2.33 a | 13.53 ± 3.01 b |
| | A | 0.01 | 11.45 ± 1.88 c | 15.18 ± 2.63 a | 12.99 ± 1.37 b | 11.18 ± 1.78 c |

*P*—probability associated with ANOVA *F*-test for tree species effect; different letters designate homogeneous groups based on Tukey's HSD post-hoc tests. O—forest floor; A—mineral horizon.

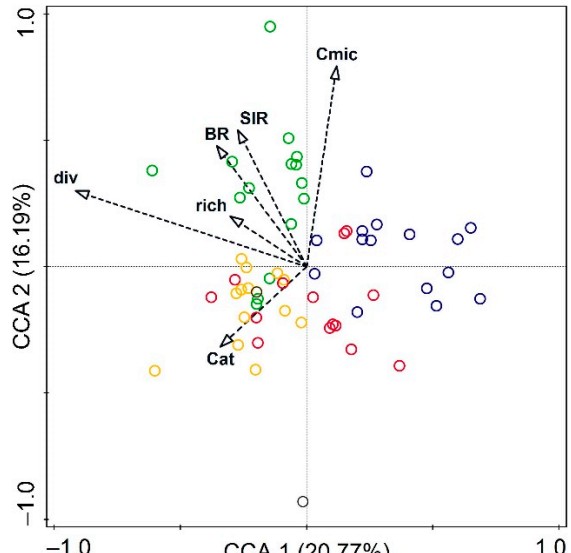 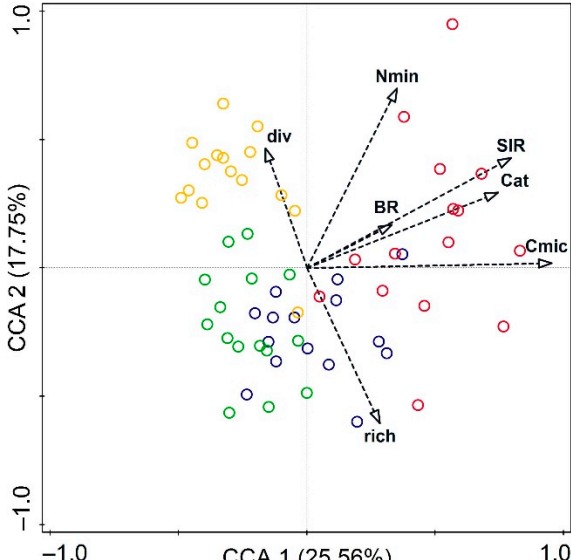

**Figure 2.** Canonical correspondence analysis for soil microbial community properties in relation to stand tree species composition within the O-horizon (**left**) and A-horizon (**right**). Arrows represent the direction of the steepest increase of individual microbial properties. Circles represent individual sampling sites, with color distinctions of dominant tree species in the stands (blue—*Pseudotsuga menziesii*; green—*Picea abies*; yellow—*Fagus sylvatica*; red—*Acer pseudoplatanus*).

Generally, the Douglas fir litter exhibited higher microbial activity, as well as richness and diversity of functional groups, than the spruce litter. Unlike the conifers, the differences in microbial characteristics (except the diversity index) between the broadleaves were negligible. Interestingly, the litter of deciduous trees exhibited lower microbial biomass than the litter of conifers. The highest microbial biomass activity, as well as richness and diversity of microbial functional groups, were typical for the litter in the Douglas fir stands.

While, in the O-horizon, the Douglas fir litter differed significantly from the others in microbial characteristics, in the A-horizon the situation was quite different and, generally, no differences were found between the soils of the spruce and the Douglas fir stands (except the diversity index). On the other hand, while basal respiration and enzyme activity did not differ between the beech and the maple stands, the microbial biomass, SIR and indices of microbial community structure did. In soils under the maples, higher microbial biomass occurred, and there seemed to be a trend of higher activity as well. Within the A-horizon, N-min was the only microbial property that distinguished deciduous and coniferous forest stands.

The community structure, based on the Biolog® method, showed differences in the utilization of 25 substrates in the O-horizon, and of only 16 in the A-horizon between the evaluated forest stands (Figure 3, Table S1). The Douglas fir litter showed high utilization in the majority of substrates, followed by the litter of the beech and the maple, while the litter of the spruce generally showed the lowest utilization rates. Surprisingly, the most distinct differences in utilization were observed between the litter of two conifers. Glycogen was the only substrate for which utilization was found to be the lowest in the Douglas litter. Xylose showed different utilization between the litter of the conifers and the broadleaves, with higher intensity in coniferous stands. For the sycamore, maple and beech stands, differences in utilization were observed only in five substrates—D-xylose, α-cyclodextrin, glycogen, α-ketobutyric acid and putrescine.

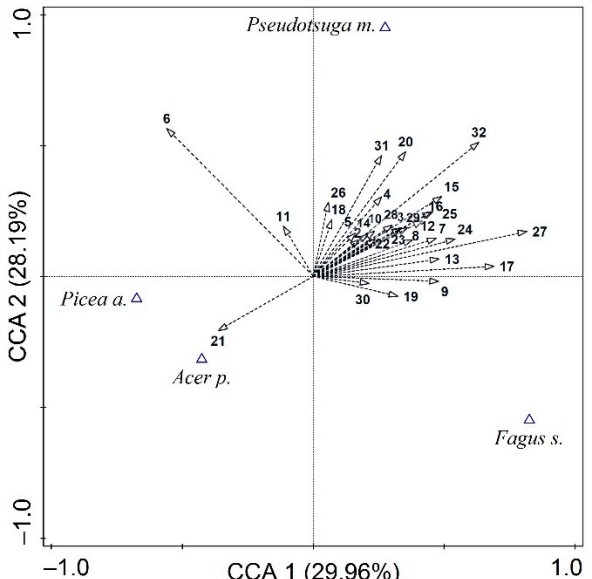 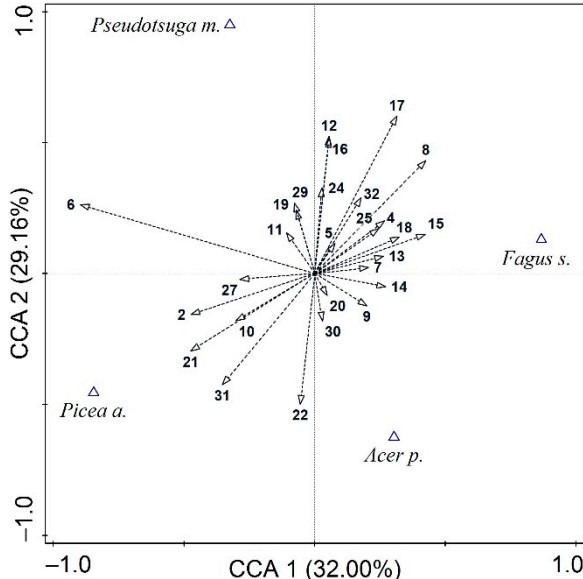

**Figure 3.** Canonical correspondence analysis based on community-level physiological profiles in different forest stands in O-horizon (**left**) and A-horizon (**right**). Arrows represent the direction of the steepest increase of activity within individual functional groups. Triangles represent individual tree species (2—β-methyl-D-glucoside; 3—D-galactonic acid γ-lactone; 4—L-arginine; 5—pyruvic acid methyl ester; 6—D-xylose; 7—D-galacturonic acid; 8—L-asparagine; 9—Tween 40; 10—i-erythritol; 11—2-hydroxy benzoic acid; 12—L-phenylalanine; 13—Tween 80; 14—D-mannitol; 15—4-hydroxy benzoic acid; 16—L-serine; 17—α-cyclodextrin; 18—N-acetyl-D-glucosamine; 19—γ-hydroxybutyric acid; 20—L-threonine; 21—glycogen; 22—D-glucosaminic acid; 23—itaconic acid; 24—glycyl-L-glutamic acid; 25—D-cellobiose; 26—glucose-1-phosphate; 27—α-ketobutyric acid; 28—phenylethyl-amine; 29—α-D-lactose; 30—D,L-α-glycerol phosphate; 31—D-malic acid; 32—putrescine).

In the A-horizon, the utilization pattern of substrates differed compared to litter. While in the Douglas fir stands, litter exhibited higher utilisation activity than the spruce litter, in the A-horizon four substrates were found (glycogen, D-glucosaminic acid, α-ketobutyric acid, D-malic acid) with higher utilization in soils under the spruce. In the soils under the beech stands, L-asparagine, α-cyclodextrin, glycyl-L-glutamic acid and α-D-lactose were more utilized, while D-xylose was less utilized than in the sycamore maple stands.

## 4. Discussion

Conversion of agricultural land to forest is known to change soil chemical properties, including carbon stocks serving as the main energy source for soil microbiota [24–26]. Shifts in vegetation cover, whether through afforestation or natural colonization of agricultural areas by trees, cause changes in carbon sequestration [27]. They also affect the storage of soil organic matter by changing the quality and quantity of litter entering the soil [28–31]. This has been the main focus of studies hitherto: the effect of tree species was mostly studied in relation to soil organic matter content, carbon sequestration and soil reaction. Several studies have shown that in forest soils more SOC was stored under coniferous trees in the upper horizons than under broadleaves, while in afforested agricultural soils SOC sequestration did not differ between broadleaf and coniferous trees [32]. Generally, soils under conifers were found to be more acid, with higher thickness of surface organic layer and C:N ratio, and less water-soluble substances leached from litter [8]. This is explained by the chemical composition of conifer litter containing more components recalcitrant to decomposition than broadleaf litter, which can result in litter accumulation on the soil surface, and the formation of acidic compounds [7]. Tree species also exert differential effects on soil fauna (microarthropods, earthworms, nematodes), while litter quality seems to be an important factor [33]. Soil fauna also mediates the effect of trees on soil prop-

erties such as layer thickness or carbon accumulation [34]. The dominant tree species have a greater effect on soil biota richness and composition than tree richness *per se* [35]. Changes in soil properties affect the function, structure and activity of the soil microbial community [36,37]. As the quality of litter and root exudates vary considerably among tree species, the effects on soil microbial community composition and microbial activity are expected to differ depending on dominant tree species, stand management and other factors, and need not necessarily be positive in terms of diversity and microbial functions [38].

Our results confirmed that a period of 45 years is long enough for the manifestation of changes in soil properties in the uppermost soil horizons triggered by the tree layer. The succession of all communities associated with trees (including soil microbiota) started in a relatively homogeneous area: pasture grasslands with small variation in the herbaceous vegetation, located within a narrow altitudinal range on a shallow slope with the same aspect, covered by the same soil type. Consequently, soil properties can reasonably be expected to have been quite similar across the area. In spite of this homogeneity, both soil physical and chemical properties and soil microbiota composition diverged during the 45 years, depending on the dominant tree species. However, the differences in successional trajectories cannot be simplified as contrasts between conifers and broadleaves. Multivariate analyses of physico-chemical soil properties indicated an overlap between the soils under the Douglas fir and the spruce, but a clear separation of beech from sycamore. Acidifying effect has frequently been attributed solely to conifers [11,39,40]. However, more recent studies contradict this assumption [14]. Soil pH and base saturation is often lower in stands dominated by beech compared to other temperate broadleaves [41,42]. This was confirmed by our study: no difference was observed in the O-horizon, and beech stands exhibited the same pH in the A-horizon as both conifers, while only the soils under the sycamore stands were less acidic, and richer in base cations. Conifers also exhibited generally higher C and N content in the O-horizon, but in the top 10 cm of mineral soil the C and N content was significantly higher only under the maple stands. Litter quality, especially with regard to the content of recalcitrant substances and decomposition rate, seems to be the main driver of nutrient cycling and soil chemical properties [42–44].

Changes in soil properties also mean changing living conditions for soil microbiota. Plant species are unique in their effects on the belowground system. Providing the matter decomposed by soil microorganisms, trees influence soil microbiota essentially in the same way as other plants, but their effect is potentially stronger because of a greater biomass [45,46]. The effect of afforestation on the composition, biomass and activity of microbial communities after afforestation is thus usually dramatic [31,37,47,48]. As in the case of physico-chemical properties, soil microbial community parameters in our study significantly differed depending on the stand-forming tree species, but again without a clear conifer-broadleaf contrast, and with different patterns in the soil horizons. In general, both microbial activity and diversity were, surprisingly, highest under the Douglas fir, followed by the sycamore, with the beech and the spruce showing mostly lower values. Utilization patterns of Biolog® substrates also differed between tree species, although by no means identically in both soil horizons. This is not surprising, as in the A-horizon the organic fraction represented only a small part of the soil mass compared to the surface organic layer, and had undergone chemical transformation.

## 5. Conclusions

Land cover in central Europe has undergone dramatic changes during the last few centuries, especially in mountainous areas. Initially, large areas were deforested to gain pastures and partly also arable land. Currently, this trend was reversed, and many former pastures were afforested or colonized by trees after being abandoned. The same applies to tree species composition of forest stands: since the 18th century Norway spruce monocultures gradually replaced natural broadleaved and mixed forests on a large scale because of its relatively fast growth and wood quality, but prolonged drought periods and climate warming during the last decades have led to increasing spruce mortality especially in

pure stands. Alternatives to spruce are currently sought also among introduced species; Douglas fir, which seems to be more tolerant to heat and drought, is considered a suitable replacement of Norway spruce in many parts of Europe [49,50]. The knowledge of the effects of tree species on soil processes is thus indispensable for planning of the composition of future forests. Objects such as the Vrchdobroč area, where stands of various species have been planted in a relatively homogeneous area and have undergone various sylvicultural treatments, are invaluable for studying the effects on soil properties and soil microbial community.

**Supplementary Materials:** The following are available online at https://www.mdpi.com/xxx/s1, Table S1: Means and results of Tukey's HSD tests for differences between tree species of microbial activity of functional groups (utilization of different carbon sources)

**Author Contributions:** Conceptualization, E.G.; methodology, E.G.; validation, E.G., R.G. and D.G.; formal analysis, M.K., J.Ž. and D.G.; investigation, E.G. and M.K.; data curation, E.G. and J.Ž.; writing—original draft preparation, R.G.; writing—review and editing, E.G. and D.G.; visualization, D.G., R.G. and J.Ž.; supervision, E.G.; project administration, E.G.; funding acquisition, E.G. All authors have read and agreed to the published version of the manuscript.

**Funding:** This research was funded by the Scientific Grant Agency of Ministry of Education, Science, Research and Sport of the Slovak Republic, project VEGA No. 1/0115/21, and the Slovak Research and Development Agency, project numbers APVV-19-0142 and APVV-19-0183.

**Informed Consent Statement:** Not applicable.

**Data Availability Statement:** The data presented in this study are available on request from the corresponding author.

**Acknowledgments:** We thank Jozef Capuliak and Terézia Dvorská for technical and laboratory assistance.

**Conflicts of Interest:** The authors declare no conflict of interest.

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
