# Peer review of "Differential Effects of Tree Species on Soil Microbiota 45 Years after Afforestation of Former Pastures"

_diversity, doi:10.3390/d14070515_

Round 1
Reviewer 1 Report
Dear Authors,
I have reviewed the paper "Differential effects of tree species on soil microbiota 45 years after afforestation of former pastures ". The aims of the paper are germane with Diversity topic. The paper is written with a moderate English level. The contribution of this paper to the scientific knowledge is moderate. In my opinion there some important flaws and I suggest the corrections in the file attached.

Author Response
I have reviewed the paper "Differential effects of tree species on soil microbiota 45 years after afforestation of former pastures". The aims of the paper are germane with Diversity topic. The paper is written with a moderate English level. The contribution of this paper to the scientific knowledge is moderate. In my opinion there some important flaws and I suggest the corrections in the file attached.
>we understand the objection of the reviewer concerning the methodology, and we tried to explain our view below in detailed responses. In the attached pdf version of the original submission, we found only the comments listed (and responded) below – five in total. We hope that we have not overlooked something.
Concerning the contribution to scientific knowledge, we of course will not pretend that we consider our manuscript to be a candidate for Nobel prize. However, we feel that it addresses an issue, which is still relevant, especially in the light of the fact that land management and land use have considerably changed in (central) Europe during the past half century: many agricultural areas have been completely abandoned or the intensity of use drastically decreased, and have been either afforested or have overgrown by forest or shrubs. These changes are reflected in the change of soils, including the biological component of soils, and our study addressed this aspect.
L77 In my opinion, the aim of the present study is very interesting but the experimental design lacks a fundamental part: the control site
>please see our response to the Results chapter (l. 159) below, which addresses in fact the same issue, namely the missing control.
L84-88 in my opinion appears to be of fundamental importance specify the management of these areas over the past 45 years
>we added a short description of the hitherto management of forest stands in the area. However, it is quite general, we could here only take information from general descriptions in research reports, as forest management plans defining silvicultural measures for particular compartments are not archived.
L118+144 please, check the formatting of references
>we apologize for this mistake – formatting was corrected and references were renumbered.
L159 the results are very interesting and significant, but without a comparison with the control site are of little scientific interest. In fact, we cannot know if the afforestation has changed and how the previous agricultural soil
>we suppose that the missing control is understood as samples from a pasture. Of course, inclusion of such samples would be useful and would add another dimension to our study. The problem is that the control does not exist. The afforested area is practically contiguous, it does not contain any pastures. There are non-afforested fragments, but are not used as pastures, and are overgrown by diverse mixtures of grasses and other herbs, frequently ruderals. Pasture grasslands, when abandoned, also underlie secondary succession, plant species composition changes. Senior authors of this manuscript participated in a study done on a proximate locality focusing on secondary succession on abandoned grassland (Janišová et al. 2007 Ann. Bot. Fennici 44: 256–266); it revealed that those parts, which were not colonized by forest, overgrew by expansive clonal grasses such as Avenula adsurgens or Calamagrostis sp. So even if we had non-afforested areas available, they would not be identical with the status 45 years ago. There are still pastures in the vicinity, but their management differs (cows instead of sheep, small private farms instead of collectivized agriculture during the communism), and it is questionable whether site conditions are identical, as they are located at lower elevations and on steeper slopes.
However, the need for such a control depends from the question asked. We were not so much interested in the difference between agricultural and forest soil. Our question was, whether different tree species induced different changes in the organic horizon and the uppermost part of the mineral soil in terms of physico-chemical properties and microbial community. We observed such differences. Just theoretically, soil under one tree species may have remained unchanged (compared to pasture), but if the other species differed, then the soil below them must have changed; this is quite clear also without a comparison to pasture soil. Our primary interest was the effect of different tree species, and we consider an object such as Vrchdobroc as ideal for this purpose: the initial status was identical (which is never guaranteed in afforestation in forest areas) and the size of the area is small enough to guarantee also the homogeneity of climate and soil.
In any case, the absence of pasture samples is something that cannot be mended at this stage. We leave the editor the judgement, whether this aspect is prohibitive in terms of publication of our study; we are convinced, that our problem-setting does not require such control.
Reviewer 2 Report
The relationship between plant and soil physicochemical properties - microorganisms has become a frequent subject of research in recent years. The presented work follows this research trend. It certainly contributes to the broadening of knowledge on the complex relationships in forest ecosystems. The results of the presented research are also important in predicting the effects of forest management. It would therefore be worth paying a little more attention to the description of the methods:
Line 93: I understand that mineral soil samples were also taken using the 0.2x0.2 frame. It would be worth describing in more detail the method of sampling.
Line 100: Was the total sample divided according to the frame area, or was a comparable soil weight taken?
It is true that the manuscript concerns microorganisms, but in the discussion it would be worth mentioning the possible impact on microorganisms also of soil fauna - one or two sentences would suffice. However, the manuscript is still valuable.
Author Response
The relationship between plant and soil physicochemical properties - microorganisms has become a frequent subject of research in recent years. The presented work follows this research trend. It certainly contributes to the broadening of knowledge on the complex relationships in forest ecosystems. The results of the presented research are also important in predicting the effects of forest management. It would therefore be worth paying a little more attention to the description of the methods:
>We would like to thank the reviewer for the positive attitude to the manuscript and the comments.
Line 93: I understand that mineral soil samples were also taken using the 0.2x0.2 frame. It would be worth describing in more detail the method of sampling.
> Please see our response to this comment below, which addresses the same issue
Line 100: Was the total sample divided according to the frame area, or was a comparable soil weight taken?
>We added extended information into the manuscript to better explain soil sampling procedure. Samples from the O-horizon were collected using a 0.2 × 0.2 m frame put on the soil surface, while the humus layer underneath the template was cut from the surrounding by knife. Then after removing the surface organic layer, we took mineral soil samples from the depth of 0-0.1 m using the knife and shovel with a depth scale indication. We could not use a probe sampler because soil exhibited higher skeleton content. We have taken approx. 400 g of samples into plastic bags from each plot, i.e. a comparable soil weight from all sampling plots.
It is true that the manuscript concerns microorganisms, but in the discussion it would be worth mentioning the possible impact on microorganisms also of soil fauna - one or two sentences would suffice. However, the manuscript is still valuable.
>we are not sure whether we properly understood this comment – does the reviewer mean the impact of the overstorey trees on soil fauna? If so, we added a short text on this topic.
Round 2
Reviewer 1 Report
Dear Authors,the paper with the changes made is improved.Experimental design coul have been better